# In Vitro Evaluation of the Most Active Probiotic Strains Able to Improve the Intestinal Barrier Functions and to Prevent Inflammatory Diseases of the Gastrointestinal System

**DOI:** 10.3390/biomedicines11030865

**Published:** 2023-03-12

**Authors:** Alessandra Fusco, Vittoria Savio, Donatella Cimini, Sergio D’Ambrosio, Adriana Chiaromonte, Chiara Schiraldi, Giovanna Donnarumma

**Affiliations:** Department of Experimental Medicine, University of Campania “Luigi Vanvitelli”, 80138 Naples, Italy

**Keywords:** intestinal barrier, lactobacilli, innate immunity, *Salmonella* Typhimurium, EIEC

## Abstract

*Background*: The integrity of the intestinal barrier is fundamental to gut health and homeostasis; its damage can increase intestinal permeability, with translocation of bacteria and/or endotoxins from gut, and the onset of various intestinal diseases. *Lactobacillus* spp. is one of the most common probiotics normally found in fermented foods and dairy products and is known for its anti-inflammatory and immunomodulatory properties and for its ability to protect and enhance the intestinal barrier functions. The aim of this work was to evaluate the ability of different strains of *Lactobacillus* spp. to improve in vitro the integrity of the intestinal barrier, to exert anti-inflammatory and immunomodulatory activity and to prevent *Salmonella* Typhimurium and enteroinvasive *Escherichia coli* (EIEC) infections. *Methods*: We analyzed the cellular expression of tight junctions, antimicrobial peptide HBD-2, pro-inflammatory cytokines and the inhibition of pathogens adhesion and invasion in a model of co-cultured epithelial cells treated with *Lactobacillus* spp. *Results*: *L. brevis, L. reuteri* and *L. rhamnosus* proved to be more effective in protecting the intestinal epithelium. *Conclusions:* These in vitro studies can help select strains particularly active in their intended use to obtain consortia formulations that can have as much maximum yield as possible in terms of patient benefit.

## 1. Introduction

The gastrointestinal tract is defined as the most important and largest immune organ of the human body as it is the site of a close cross-talk between the cells that make up the epithelium and the immune system that regulates the contents of the intestinal lumen.

The intestinal surface is composed by a single layer of epithelial cells that form the broader interface between the host and the environment, by maintaining the balance between the secretion of fluids and the absorption of nutrients and ions, and protects it from microorganisms, toxins and dietary antigens [1,2,3].

The integrity of the intestinal barrier is fundamental for gut health and homeostasis; there are several factors that contribute to barrier integrity, such as gastric acidity, peristalsis, secreted antimicrobial factors and the mucous coat [4], produced by the activity of goblet cells, dispersed among the intestinal epithelial cells, that are responsible for secreting glycoproteins called mucins [4,5].

In addition, a complex system of intercellular junctions regulates intestinal permeability, including weak junctions (i.e., Adherens and Gap junctions, Desmosomes) and protein complexes formed at the most apical areas of polarized epithelial cells called “tight junctions” (TJs). The TJs, such as occludin, claudin and zonulin, are dynamic structures that regulate the traffic of water, solutes and immune cells from the intestinal lumen to the subepithelial tissues in both physiological and pathological conditions, whose assembly and organization can vary depending on the intracellular and extracellular stimuli received [6,7]. 

Various factors can reduce the barrier function [8,9], by compromising the integrity of the intestinal epithelium, leading to an increase in permeability with the consequent translocation of bacteria and/or their toxic products from the gut, and induction of the systemic inflammatory response. These events represent the main cause of the pathogenesis of most intestinal diseases, such as irritable bowel syndrome (IBS), inflammatory bowel disease (IBD), celiac disease and the early stages of colon cancer [7,10,11,12,13]. 

Among these factors, there are oxidative stress, pro-inflammatory cytokines and pathogenic microorganisms such as *Salmonella* Typhimurium [3,14] and enteroinvasive *Escherichia coli* (EIEC) [15]

*Salmonella enterica* subsp. *enterica* serovar Typhimurium (*S.* Typhimurium) is Gram-negative bacillus, considered a main cause of acute foodborne infection caused by non-typhoidal *Salmonella* (NTS) [14]. *S*. Typhimurium is able to colonize the intestinal epithelium by competing with the components of the resident microbiota [16], and thanks to its ability to survive inside macrophages [17], it induces the activation of the nuclear transcriptional factor NF-kB, resulting in the secretion of pro-inflammatory cytokines, such as interleukin (IL)-8 [18] and tumor necrosis factor-alpha (TNF-α) [19]. This inflammatory state causes severe symptoms such as vomiting, diarrhea, fever and, in some cases, especially in immunocompromised, children and elderly patients, death [14,15,16,17,18,19,20].

EIEC is another Gram-negative bacillus involved in the pathogenesis of bacillary dysentery that, thanks to a repertoire of virulence genes grouped into a 230 kb virulence plasmid, invades the colonic mucosa of the human hosts and destroys the intestinal epithelial barrier, causing the formation of abscesses and colonic ulceration. The resulting acute inflammatory response is responsible for a pathologic condition similar to shigellosis, characterized by the presence of purulent stools with blood and leucocytes accompanied by abdominal cramps, fever and sometimes vomiting [15,21], and can be found in both children and adults, particularly in low- and middle-income countries and inadequate sanitation conditions [15,22]. 

It was demonstrated that the intake of probiotic bacteria contributes to a proper intestinal functioning by maintaining the epithelial permeability, enhancing the mucous layer, increasing enterocyte turnover, stimulating the innate and adaptative immune response and restoring gut microbiota composition and activity [23,24].

Probiotics, according to the World Health Organization, are defined as “live microorganisms which when administered in adequate amounts confer a health benefit on the host” [25] and can enter the host through food or supplements and colonize the intestinal mucosa by being able to resist the action of gastric and biliary juices.

The genus *Lactobacillus* is one of the most widely used probiotics that can be found in a large variety of functional foods and biotherapeutic agents [26].

The genus *Lactobacillus* spp. includes a wide range of Gram-positive, facultative anaerobic and asporogenic bacteria with known beneficial effects [27]; they are naturally present in the human intestinal tract, and their distribution is heterogeneous and depends on various exogenous and endogenous factors [28]. The enhancement of epithelial barrier function and the ability to regulate both innate and adaptive immunity are some of the proposed mechanisms by which certain lactobacilli may confer beneficial activities [29]; these activities may depend on the *Lactobacillus* strain and species as well as the target population and the intestinal mucosa resilience capacity, and can play a role in disease prevention and treatment in the host through immune stimulation and regulation. Extensive studies have shown that lactobacilli have the capacity to modulate the expression of pro- and anti-inflammatory cytokines [30,31,32] and the innate immune response [33]. Among the innate immune response mechanisms, of particular interest is the production of the antimicrobial peptides (AMPs), including human beta-defensin 2 (HBD-2), an inducible AMP, identified in psoriatic lesions as the most abundant AMP, produced by numerous epithelia and with a broad spectrum of action against Gram-positive and Gram-negative bacteria, fungi and the envelopes of some viruses. Its production is induced following various stimuli, such as inflammation, infections, endogenous stimuli or wounds [3,34]. On the other hand, AMPs can also modulate the microbiota composition (unlike the antibiotics commonly administered in therapy) and stimulate the renewal of the intestinal epithelium [35].

With the growing interest of the scientific community in the use of probiotics in a variety of potential applications, the aim of this work was to evaluate the anti-inflammatory and immunomodulatory properties and the ability of different strains of *Lactobacillus* spp. to reinforce in vitro the integrity of the intestinal barrier and to inhibit adhesion and invasion of *S. tyhimurium* and EIEC in a model of co-cultured epithelial cells, in order to highlight the strains, to be used individually or in consortia that are more suitable for this purpose.

## 2. Materials and Methods

### 2.1. Cell Cultures

Human Caucasian colon adenocarcinoma Caco-2 cells (ATCC^®^ HTB-37™) and human colon epithelial mucus-secreting HT29-MTX cells (Sigma Aldrich) were first cultured separately, both using DMEM (Gibco) medium supplemented with 10% fetal bovine serum (Gibco), 100 IU/mL penicillin, 100 mg/mL streptomycin (Gibco) and 2 mM glutamine (Gibco), at 37 °C in a 5% CO_2_ atmosphere. Then, the co-cultures were set up in 12-well Transwell^®^ plates (with 12 mm polycarbonate insert and 0.4 μm pores) by plating the cells in the basolateral compartment in a 75:25 ratio Caco-2:HT29-MTX, and carried out for 21 days, changing the medium every 2 days. 

### 2.2. Bacterial Strains

*Lactobacillus* strains, including *Limosilactobacillus fermentum* isolated from buffalo milk [32], *Levilactobacillus brevis* SP-48, *Lacticaseibacillus rhamnosus* IMC501, *Limosilactobacillus reuteri* LR92 and *Lacticaseibacillus paracasei* IMC502 provided by the Incube project leader, R&D—IBSA Farmaceutici Italia, were cultivated in bioreactor fermentation processes and freeze-dried to obtain the powders used as the starting material. Before the experiment, the powders were resuspended and grown in Man, Rogosa, and Sharpe broth (MRS-Oxoid) at 37 °C in microaerophilic conditions for 24 h.

*S. Typhimurium* (ATCC^®^ 14028GFP^™^) and EIEC (ATCC^®^ 43893^™^) were cultured in Luria-Bertani broth (Oxoid; Unipath, Basingstoke, UK) at 37 °C for 18 h.

### 2.3. Cell Treatment

Co-cultured cells were treated with different strains of *Lactobacillus* spp. (10^8^ CFUs/mL) at a multiplicity of infection (MOI) of 100, alone or in combination with 20 μg/mL of lipopolysaccharide (LPS) of *S.* Typhimurium, for 24 h at 37 °C at 5% CO_2_ in DMEM without antibiotics.

### 2.4. Real-Time PCR

In order to evaluate the expression of pro- and anti-inflammatory cytokines, HBD-2, and TJs, the cells at the end of treatments were washed three times with sterile PBS, and the total RNA was extracted using High Pure RNA Isolation Kit (Roche Diagnostics). 

A total of 200 ng of cellular RNA were reverse-transcribed (Expand Reverse Transcriptase, Roche) into complementary DNA (cDNA) using random hexamer primers (random hexamers, Roche) at 42 °C for 45 min, according to the manufacturer’s instructions. Real-time PCR for *IL-6*, *IL-8*, *TNF-α*, *IL-1α*, *TGF-β*, HBD-2, *Occludin*, *Zonulin*-*1*, and *Claudin-1* was carried out with the LC Fast Start DNA Master SYBR Green kit (Roche Diagnostics) using 2 µL of cDNA, corresponding to 10 ng of total RNA in a 20 μL final volume, 3 mM MgCl_2_ and 0.5 μM sense and antisense primers (Table 1). After amplification, the melting curve analysis was performed by heating to 95 °C for 15 s at a temperature transition rate of 20 °C/s, cooling to 60 °C for 15 s with a temperature transition rate of 20 °C/s, and then heating the sample at 0.1 °C/s to 95 °C. The results were then analyzed using LightCycler software (Roche Diagnostics). The standard curve of each primer pair was established with serial dilutions of cDNA. All PCRs were run in triplicate. The specificity of the amplification products was verified using electrophoresis on a 2% agarose gel and visualization by ethidium bromide staining [6].

### 2.5. ELISA Assay

The presence of IL-6, IL-8, IL-1α, TNF-α, TGF-β, HBD-2, Zonulin-1, Occludin, and Claudin-1 in the cellular lysates of Caco-2/HT29-MTX co-cultures infected with different strains of lactobacilli with or without LPS was analyzed using enzyme-linked immunosorbent assay (ELISA; Invitrogen, IL-6, IL-8, IL-1 alpha, TNF alpha, and TGF beta-1 Human ELISA Kit; Phoenix Pharmaceuticals, Inc. Defensin 2, beta (Human)—ELISA Kit; Elabscience Biotechnology Inc. Human Zonulin, Human OCLN(Occludin) and Human CLDN1 (Claudin 1) ELISA Kit).

### 2.6. Adhesion and Invasiveness Assay

The ability of *L. brevis*, *L. reuteri*, and *L. rhamnosus* to reduce the adhesiveness and invasiveness ability of *S.* Typhimurium and EIEC in intestinal cocultures was investigated in three different experimental types: (i) competitive assay, in which intestinal epithelial cells were incubated simultaneously with lactobacilli (10^8^ CFUs/mL) and *S.* Typhimurium or EIEC (10^8^ CFUs/mL) for 2 h. (ii) Inhibition assay, in which cells were preincubated with lactobacilli (10^8^ CFUs/mL) for 2 h and then *S.* Typhimurium or EIEC (10^8^ CFUs/mL) was added and incubated for an additional 2 h. (iii) Displacement assay in which cells were pre-incubated with *S.* Typhimurium or EIEC (10^8^ CFUs/mL) for 2 h and then lactobacilli (10^8^ CFUs/mL) were added and further incubated for 2 h. At the end of this time, the infected monolayers were extensively washed in PBS, then lysed with a solution of 0.1% Triton X-100 (Sigma-Aldrich, St. Louis, MO, USA) in PBS for 10 min at room temperature to count the internalized bacteria. The aliquots of cell lysates were serially diluted and plated on Hektoen agar (OXOID) and incubated at 37 °C overnight to quantify the total viable cell-associated bacteria (CFUs/mL). For invasiveness assays, after the incubation with bacteria, infected monolayers were extensively washed with sterile PBS and further incubated for an additional 2 h in the DMEM medium, supplemented with gentamicin sulphate (250 μg ml^–1^) (Sigma-Aldrich) in order to kill the extracellular bacteria, then cells were lysed and plated as previously described to quantify the total number of internalized bacteria (CFUs/mL).

### 2.7. Statistical Analysis

Significant differences among groups were assessed through two-way ANOVA using GraphPad Prism 8.0, and the comparison between the means was calculated using a Student’s *t*-test. The data are expressed as means ± standard deviation (SD) of three independent experiments.

## 3. Results

### 3.1. Regulation of TJ Expression

The maintenance of the integrity of the intestinal barrier is essential to ensure the osmotic balance and to protect the host from the translocation of pathogens. For this reason, it is important that the junctional protein complexes are functional and regularly expressed. For this purpose, the cell co-cultures were treated, following their differentiation, with the different lactobacilli strains, to evaluate their ability to strengthen the integrity of the epithelium. The data show that the expression levels of the *TJs* genes (Figure 1A) and their corresponding proteins (Figure 1B) are significantly induced in the presence of all *Lactobacillus* strains, especially with *L. brevis* and *L. rhamnosus*.

### 3.2. Induction of Innate Immune Response

The induction of HBD-2 production by intestinal epithelial cells by the different *Lactobacillus* strains was evaluated using real-time PCR and ELISA assay. As shown in Figure 2A,B, the AMP is induced in the presence of lactobacilli, but the most significant induction, unlike in the case of TJs, occurs in the presence of *L. reuteri*.

### 3.3. Anti-Inflammarory Activity of Lactobacillus spp.

The co-cultures were also used as an experimental model to evaluate the ability of lactobacilli to reduce the cellular inflammatory response following treatment with *S.* Typhimurium LPS. The results obtained (Figure 3A,B) show that the expression levels of pro-inflammatory cytokines and their corresponding proteins are constantly and significantly reduced in the presence of all strains of lactobacilli, with the exception of *L. fermentum* and *L. paracasei*, which do not seem to be involved in the modulation of IL-8.

### 3.4. Activity of Lactobacilli against S. Typhimurium and EIEC Adhesion and Invasion

The strains of lactobacilli that showed greater efficacy in previous analyses, in particular *L. reuteri*, *L. rhamnosus*, and *L. brevis*, were tested for their ability to interfere with the adhesion and invasion ability of EIEC and *S*. Typhimurium. The assays were performed in three different ways, inhibition, competition, and displacement. The results obtained (Figure 4) show that for EIEC, all three strains were able to significantly reduce both adhesion and invasion. Considering the adhesion tests, the main results occurred during the competition assay (EIEC adhesion control ~3.5 × 10^8^ CFUs/mL), while in the invasion experiment both the inhibition and competition assay were effective (EIEC invasion control ~4 × 10^4^ CFUs/mL).; for *S*. Typhimurium, all three strains were able to significantly reduce the adhesion both in the inhibition and in the competition assay (*S*. Typhimurium adhesion control ~10^9^ CFUs/mL), while they had no effect on the invasiveness (*S*. Typhimurium invasion control ~ 3 × 10^3^ CFUs/mL).

For both pathogenic strains, the displacement assay had no significant effect.

## 4. Discussion

Maintaining the integrity of the intestinal barrier is of crucial importance in the prevention of the onset not only of intestinal diseases, but also of metabolic, autoimmune, and nervous system disorders [36,37]. The numerous studies carried out on the intestinal mucosa have shown that it protects the host through the stimulation of the mechanisms of innate and acquired immunity [38], and guarantees homeostasis thanks to the maintenance of the microbiota balance [39].

Lactobacilli are probiotics commonly found in fermented foods and in the gut microbiota of humans and animals [40,41]. In recent years, great progress was achieved in the study of the mechanisms of symbiosis between lactic acid bacteria and the host. Their main beneficial effects have been shown to include the regulation of the imbalance of the intestinal flora [42], the strengthening of the intestinal barrier functions [23,27,36], the maintenance of homeostasis [36,43], the regulation of the immune system [43], and the production of neurotransmitters (gut–brain axis) [44]. It has been also shown that the exopolysaccharide (EPS) produced by lactobacilli has the peculiar ability to modify the microbiota [45], and to improve the colonization and growth of intestinal bacteria by acting as a carbon source [46]. Furthermore, probiotics have the ability to compete with the colonizing pathogens of the gastrointestinal tract, preventing their adhesion, and inhibiting their growth due to the lowering of the pH caused by lactic acid production [36]. 

In vitro models simulating the human colon represent a useful tool for mechanistic studies on the interactions of probiotics with the intestinal epithelium [47]; in this work, we used an experimental model of co-cultures of intestinal epithelial cells, Caco-2 and HT29-MTX, and we analyzed their interaction with different strains of *Lactobacillus* spp. in order to determine which of these strains showed the best functionality protecting the intestinal epithelium, by strengthening the barrier, and promoting anti-inflammatory and immunomodulating properties.

The choice to work with a co-culture was due to several pieces of evidence: first of all, the transepithelial electrical resistance (TEER) is much higher in Caco-2 monolayers (up to 500 ohm cm^2^) than in the human intestine (12–69 ohm cm^2^) because of a high expression of TJs; in addition, the intestinal barrier is composed of several cellular phenotypes, including enterocytes, goblet cells, Paneth cells, endocrine cells, and stem cells. The co-culture model that combines the two major cellular phenotypes found in the gut, Caco-2 and mucus-secreting HT29-MTX, provides a mucus-coated epithelial monolayer that most efficiently mimics the condition occurring *in vivo*. The effectiveness of this model has been demonstrated by Mahler et al. [48].

The choice of the strains was due to them being commercialized and having already been the subject of various studies; in fact, *L. brevis* SP-48 is present in commercial products (e.g., Florap lady) that promote the balance of the intestinal flora and the functionality of the urinary tract. Moreover, our recent study indicated that it inhibited *H. pylori* and acted as modulator of the immune system, reducing the inflammation potentially related to the treatment of intestinal bowel disease [49].

Similarly, the activity of *L. fermentum* was evaluated in a gastric epithelial cell model demonstrating the modulation of inflammatory cytokines and the inhibition of *H. pylori* [32]; moreover, other *L. fermentum* strains showed anti-infectious and immunomodulatory properties [50].

*Lactobacillus rhamnosus* IMC 501 and *Lactobacillus paracasei* IMC 502 are commercially available probiotic strains (e.g., SYNBIO). Verdenelli and co-authors demonstrated persistence of the two strains, administered in combination, in the intestinal tract of test subjects, and an improvement in the natural regularity and intestinal well-being [51,52].

*L. brevis* normally colonizes the human gastrointestinal tract and it has been shown to provide several health-stimulating effects [53], some of which enhance the mucosal barrier function [54]. *L. reuteri* LR92 is also present in commercial products and clinical studies demonstrated that it improves colic symptoms in new-borns if preventively administered to mothers in the last months of pregnancy [55]. 

In the first part of the work, we therefore treated the co-cultures, which were differentiated for 21 days, with the different strains of *Lactobacillus* spp. for 24 h, and we analyzed, by real-time PCR and ELISA assay, the expression levels of the TJs Occludin, Zonulin-1, and Claudin-1, and the antimicrobial peptide HBD-2. The results obtained revealed that all the lactobacilli strains tested have the ability to significantly induce the expression of TJs, in particular *L. brevis* and *L. rhamnosus*. Regarding HBD-2, on the other hand, a significant induction of peptide production is appreciated especially in the presence of *L. reuteri.*

In order to evaluate the anti-inflammatory activity of lactobacilli, the co-cultures were treated for 24 h with *S.* Typhimurium LPS alone or in combination with different strains, and the expression levels of pro- and anti-inflammatory cytokines were evaluated using real-time PCR and ELISA assay. The data obtained revealed that, except for IL-8 whose expression was apparently not modulated by *L. fermentum* and *L. paracasei*, all the strains could significantly induce the expression of pro-inflammatory cytokines and induce that of the anti-inflammatory cytokine TGF-β.

On the basis of the data obtained, and considering that all the probiotic strains analyzed showed a significant anti-inflammatory activity, we can state that three specific strains seem to have a particularly beneficial effect on the intestinal mucosa: *L. reuteri*, which emerges for its ability to significantly stimulate the immune defenses by promoting the massive release of HBD-2, *L. rhamnosus*, and *L. brevis*, which revealed a high ability to strengthen the intestinal barrier, a necessary condition for the correct maintenance of various functions, including the absorption of micronutrients and the maintenance of homeostasis. 

In the second part of the work, we therefore evaluated the ability of these three strains to inhibit the adhesion and invasion of *S.* Typhimurium and EIEC in the intestinal mucosa.

For this purpose, adhesion and invasion assays were carried out in three different ways—inhibition, competition, and displacement—to understand the dynamics of the interaction between the lactobacilli and the pathogens. In fact, by preincubating with lactobacilli, their ability to produce bacteriocins and to lower the pH of the environment can exert a killing activity that inhibits the adhesion of the pathogen; on the other hand, the simultaneous addition of both species could cause a competition between the two bacterial species, both for binding to the cellular receptor sites and for uptake of the nutritive factors; finally, the addition of the probiotic following the adhesion of the pathogen can cause its displacement from the binding sites, also due to the killing activity exerted by the bacteriocins. 

The results obtained showed that all three strains significantly inhibited EIEC adhesion in competition assays, and blocked its invasion through competition and inhibition. This different behavior could be due to the fact that lactobacilli are not able to prevent the adhesion of the pathogen if they are taken preventively, while they can compete with it; however, once adhesion has occurred, the barrier integrity-enhancing activity of the lactobacilli (which increase tight-junction expression) prevents the invasion of the intestinal submucosa. As far as *S.* Typhimurium is concerned, however, the inhibition activity is mainly carried out in the adhesion phase, both in the inhibition and in the competition assay, probably because it avoids upstream the adhesion of the pathogen, thereby reducing significantly the bacterial load that invades the intestinal submucosa, which is already significantly reduced. In the displacement assays, satisfactory results were not obtained for both strains, neither regarding adhesion nor invasion, demonstrating that the lactobacilli are evidently not able to exert their protective action against the two pathogens if taken when the infection has already occurred.

## 5. Conclusions

Although the use of probiotic-based supplements has been practiced for many years, lately the interest in this type of product significantly increased because of the growing public interest in the use of natural benefits/resources [56].

It was widely demonstrated that lactobacilli are probiotics that can contribute to the maintenance of the essential functions for an optimal state of health; however, these in vitro studies show that different probiotic strains can act differently with respect to the epithelium and towards the pathogen with which they interact; for this reason it would be useful, when designing probiotic-based formulations, to take into consideration the specificity of the strains and their intended use, so as to obtain consortia formulations able to achieve the maximum yield in terms of patient benefit.

## Figures and Tables

**Figure 1 biomedicines-11-00865-f001:**
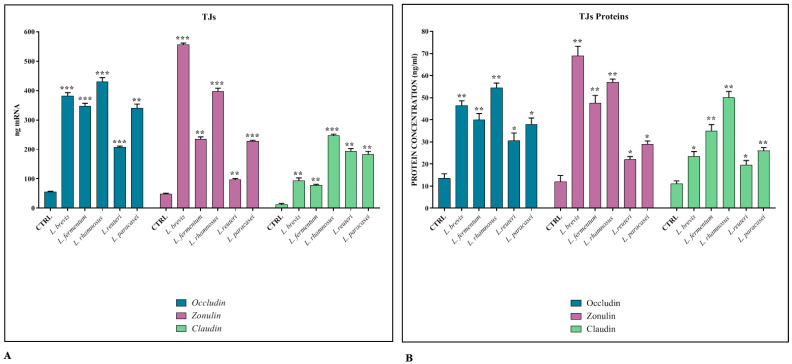
TJ expression. Comparison between relative gene expression (**A**) and protein concentration (**B**) in Caco-2-HT29-MTX co-cultures treated with different strains of *Lactobacillus* spp. The data are representative of three different experiments ± SD. Significant differences are indicated by * *p* < 0.05, ** *p* < 0.01, *** *p* < 0.001.

**Figure 2 biomedicines-11-00865-f002:**
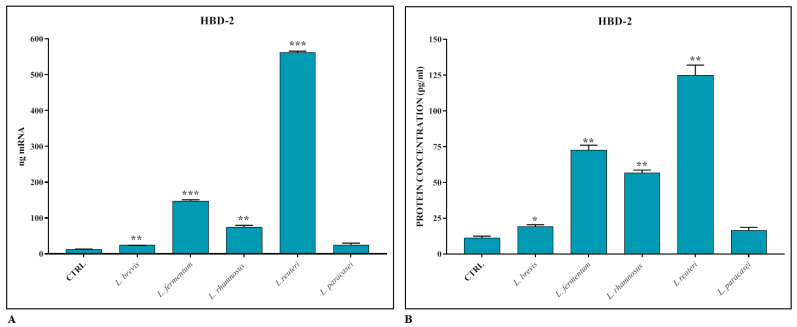
Gene expression (**A**) and protein concentration (**B**) of HBD-2 in Caco-2-HT29-MTX co-cultures treated with different strains of *Lactobacillus* spp. The data are representative of three different experiments ± SD. Significant differences are indicated by * *p* < 0.05, ** *p* < 0.01, *** *p* < 0.001.

**Figure 3 biomedicines-11-00865-f003:**
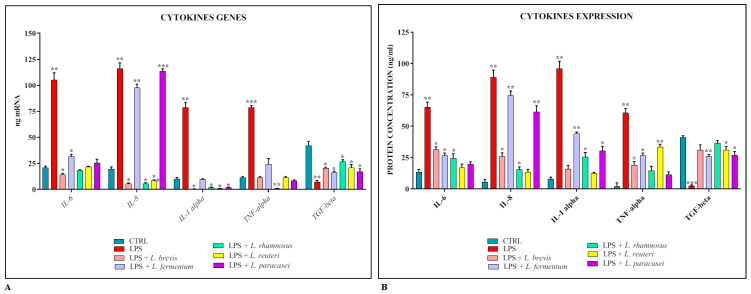
Comparison between relative gene expression (**A**) and protein concentration (**B**) in Caco-2-HT29-MTX co-cultures treated with LPS of *S.* Typhimurium with or without *Lactobacillus* spp. The data are representative of three different experiments ± SD. Significant differences are indicated by * *p* < 0.05, ** *p* < 0.01, *** *p* < 0.001.

**Figure 4 biomedicines-11-00865-f004:**
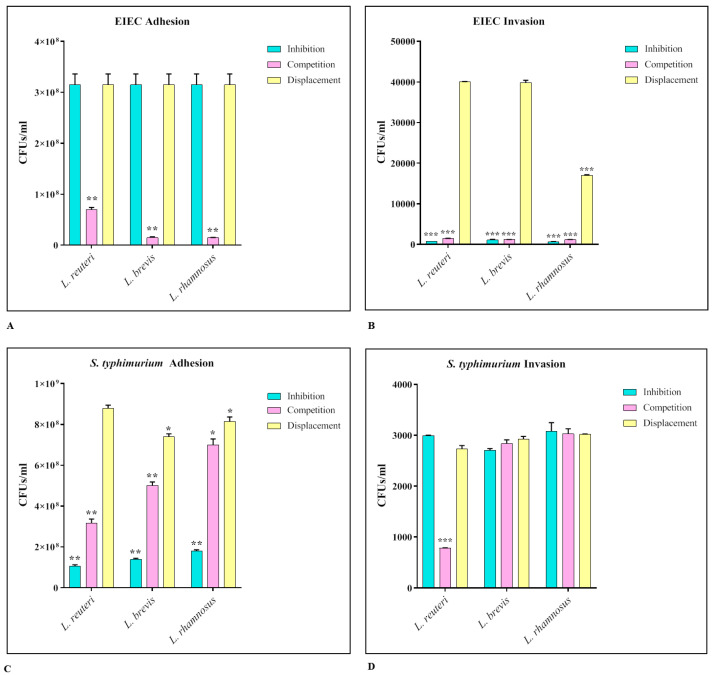
EIEC (**A**,**B**) and *S.* Typhimurium (**C**,**D**) adhesion and invasion assays. The number of live cell-associated bacteria was determined by host cell lysis, plating, and counting of CFUs/mL. The data are representative of three different experiments ± SD. Significant differences are indicated by * *p* < 0.05, ** *p* < 0.01, *** *p* < 0.001.

**Table 1 biomedicines-11-00865-t001:** Primer sequences and amplification programs.

Gene	Primer Sequences	Conditions	Product Size (bp)
*IL-6*	5′-ATGAACTCCTTCTCCACAAGCGC-3′ 5′-GAAGAGCCCTCAGGCTGGACTG-3′	5″ at 95 °C, 13″ at 56 °C, 25″ at 72 °C for 40 cycles	628
*IL-8*	5′-ATGACTTCCAAGCTGGCCGTG-3′ 5′-TGAATTCTCAGCCCTCTTCAAAAACTTCTC-3′	5″ at 94 °C, 6″ at 55 °C, 12″ at 72 °C for 40 cycles	297
*IL-1α*	5′-CATGTCAAATTTCACTGCTTCATCC-3′ 5′-GTCTCTGAATCAGAAATCCTTCTATC-3′	5″ at 95 °C, 8″at 55 °C, 17″ at 72 °C for 45 cycles	421
*TNF-α*	5′-CAGAGGGAAGAGTTCCCCAG-3′ 5′-CCTTGGTCTGGTAGGAGACG-3′	5″ at 95 °C, 6″ at 57 °C, 13″ at 72 °C for 40 cycles	324
*TGF-β*	5′-CCGACTACTACGCCAAGGAGGTCAC-3′ 5′-AGGCCGGTTCATGCCATGAATGGTG-3′	5″ at 94 °C, 9″ at 60 °C, 18″ at 72 °C for 40 cycles	439
*HBD-2*	5′-GGATCCATGGGTATAGGCGATCCTGTTA-3′ 5′-AAGCTTCTCTGATGAGGGAGCCCTTTCT-3′	5″ at 94 °C, 6″ at 63 °C, 10″ at 72 °C for 50 cycles	198
*Occludin*	5′-TCAGGGAATATCCACCTATCACTTCAG-3′ 5′-CATCAGCAGCAGCCATGTACTCTTCAC-3′	10″ at 95 °C, 45″ at 60 °C for 40 cycles	188
*Zonulin-1*	5′-AGGGGCAGTGGTGGTTTTCTGTTCTTTC-3′ 5′-GCAGAGGTCAAAGTTCAAGGCTCAAGAGG-3′	10″ at 95 °C, 45″ at 60 °C for 40 cycles	217
*Claudin-1*	5′-CTGGGAGGTGCCCTACTTTG-3′ 5′-ACACGTAGTCTTTCCCGCTG-3′	1″ at 95 °C, 30″ at 60 °C, 20″at 72 °C for 40 cycles	128

## Data Availability

The authors confirm that the data supporting the findings of this study are available within the article.

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
