# Peer review of "In Vitro Evaluation of the Most Active Probiotic Strains Able to Improve the Intestinal Barrier Functions and to Prevent Inflammatory Diseases of the Gastrointestinal System"

_biomedicines, 2023, doi:10.3390/biomedicines11030865_

Round 1

Reviewer 1 Report

This original article addresses a subject of interest: the screening of effective probiotic strains. The topic is relevant for the field as the impact of probiotics on human health has been a theme of intensive research for several groups during recent years.

The article deserves to be published in Biomedicines, however, there are several points in which improvements should be made to strengthen the manuscript.

·       I suggest the reformulation of the title to better suit the explorative activities described in the article.

·       The aim of the study is not visible in the abstract. Pleas present it.

·       A short description of the reasons why you chose the five strains in detriment to others would be helpful for the reader.

·       Line 151 – IL-1α is absent in Table 1, although it is mentioned in line 140. Please add IL-1α in Table 1.

·       Why did you choose to perform the adhesion and invasiveness assay without a control group? 

·       Line 218 - Subtitles 3.3 and 3.4 are identical. Please revise subtitle 3.4.

·       Line 235 – Please remove 540

·       How did the five strains you studied behave in other experiments? Your results showed significant differences between strains. For comparison, please add more specific references in addition to the general lactobacilli discussion already present in the article.     

·       Line 247 – Why did you introduce the abbreviation LAB? It is mentioned only here.  

·       A dedicated section for conclusions would be beneficial for the reader. The conclusions are consistent with the evidence and arguments presented.

·       Some references are in duplicate, please revise: 6 and 36; 32 and 35. Although there are several self citations, the majority of references are appropriate.

·       Please increase the quality of the figures. The text is difficult to read.

·       Please ensure that all the names of the species are written in italics.

Author Response

This original article addresses a subject of interest: the screening of effective probiotic strains. The topic is relevant for the field as the impact of probiotics on human health has been a theme of intensive research for several groups during recent years.

The article deserves to be published in Biomedicines, however, there are several points in which improvements should be made to strengthen the manuscript.

Dear Reviewer,

thank you so much for your positive evaluation of our manuscript and for appreciating our work.

  1. I suggest the reformulation of the title to better suit the explorative activities described in the article.

As requested, we have remodulated the title making it clear that the purpose of our analysis is not only a screening but a real search for the most suitable strains in the prevention of gastrointestinal disorders.

  1. The aim of the study is not visible in the abstract. Pleas present it.

Done.

  1. A short description of the reasons why you chose the five strains in detriment to others would be helpful for the reader.

The reasons for our choice, as now specified in the manuscript (lines 358-376), depend on the fact that these strains have already been studied and are marketed by the company, mentioned in the acknowledgments, which is involved in the project that financed our study.

  1.  Line 151 – IL-1α is absent in Table 1, although it is mentioned in line 140. Please add IL-1α in Table 1.

It is true, we had mixed up with Il-1b, which is not present in this work. The error has been corrected.

  1. Why did you choose to perform the adhesion and invasiveness assay without a control group?

The control test with the pathogens alone was regularly carried out, only that it is not present on the graphs, as these are set to the 3 different modes (inhibition, competition and displacement), which obviously could not be conducted with the pathogen alone. However, the data on the controls had been entered in the results section (in the updated version of the manuscript, it is found in the lines 282-286).

  1. Line 218 - Subtitles 3.3 and 3.4 are identical. Please revise subtitle 3.4.

Done.

  1. Line 235 – Please remove 540

Done.

  1.  How did the five strains you studied behave in other experiments? Your results showed significant differences between strains. For comparison, please add more specific references in addition to the general lactobacilli discussion already present in the article.    

Of the five strains analyzed in the first part of the work, the 3 that showed greater activity against the intestinal epithelium in our in vitro system were selected. For this reason, L.fermentum and L. paracasei were not used in Salmonella and EIEC adhesion and invasion assays. This, as specified in the discussion section (lines 392-398) does not mean that they do not have a recognized anti-inflammatory activity, but only that for the other functions we are looking for (induction of the expression of TJs and of HBD-2) they appear to be less active ingredients of others, and we help demonstrate that analytical work like ours can help optimize the beneficial effects of probiotic-based formulations. More than a comparison between the various strains, therefore we have inserted in the text some bibliographic references relating specifically to all the strains analyzed in this work (lines 358-376).

  1. Line 247 – Why did you introduce the abbreviation LAB? It is mentioned only here. 

Sorry, it’s a typo that was removed.

  1. A dedicated section for conclusions would be beneficial for the reader. The conclusions are consistent with the evidence and arguments presented.

As requested, the section Conclusion was added.

  1. Some references are in duplicate, please revise: 6 and 36; 32 and 35. Although there are several self citations, the majority of references are appropriate.

The duplicates were removed and now the self-citations are 5 as consented.

  1. Please increase the quality of the figures. The text is difficult to read.

Done.

  1. Please ensure that all the names of the species are written in italics.

Done.

Reviewer 2 Report

The manuscript by A. Fusco et al. entitled "Creation of an in vitro Model for the Screening of Different Probiotic Strains Able to Improve Intestinal Barrier Functions and to Prevent Inflammatory Diseases of the Gastrointestinal System" is devoted to the  in vitro study of probiotic Lactobacillus strains on intestinal epithelial cells infected with S. Typhimurium ATCC14028 and enteroinvasive E. coli ATCC43893. The authors used qPCR, ELISA, adhesion and invasion assays. In general, the manuscript is well-written but some minor issues can be pointed out below.

Line 2: I suggest to italisize in vitro here and in the manuscript text.

Line 12: Omit the first e-mail, as the corresponding author has the email [email protected]

Abstract: Omit sections (background, methods, results, conclusions).

Lines 15, 22, 27 and further in the text: Typhimurium is the serovar name, therefore it must be not italisized and must start with a capital letter (Salmonella Typhimurium).

Line 59: I suggest adding subspecies here as follows: Salmonella enterica subsp. enterica serovar Typhimurium (S. Typhimurium)

Lines 90, 94, and further in the text: lactobacilli should not be italisized

Line 107: EIEC should not be italisized

Line 116: Subscript as CO2

Line 127: What is E.?

Line 141: Add Roche for the LC Fast Start DNA Master SYBR Green kit

Line 156: Add further details for ELISA: e.g. kit names etc.

Line 190-191: Italisize Lactobacillus, L. brevis, L. rhamnosus

Figures 1-4: Use better resolution for figures: text is not recognizable.

Line 201: Italisize L. reuteri

Figure 3: The right side of the figure doesn't fit.

Line 220: Italisize L.reuteri, L.rhamnosus and L. brevis

Line 235: What is 540?

Lines 323-324: Add Institutional Review Board Statement and Informed Consent Statement

Author Response

Reviewer 2:

The manuscript by A. Fusco et al. entitled "Creation of an in vitro Model for the Screening of Different Probiotic Strains Able to Improve Intestinal Barrier Functions and to Prevent Inflammatory Diseases of the Gastrointestinal System" is devoted to the  in vitro study of probiotic Lactobacillus strains on intestinal epithelial cells infected with S. Typhimurium ATCC14028 and enteroinvasive E. coli ATCC43893. The authors used qPCR, ELISA, adhesion and invasion assays. In general, the manuscript is well-written but some minor issues can be pointed out below.

Dear Reviewer,

thank you so much for your positive evaluation of our manuscript and for appreciating our work.

  1. Line 2: I suggest to italisize in vitro here and in the manuscript text.

Done.

  1. Line 12: Omit the first e-mail, as the corresponding author has the email [email protected]

Sorry, we have missed the asterisk on the other corresponding author Alessandra Fusco.

  1. Abstract: Omit sections (background, methods, results, conclusions).

The division of the abstract into sections falls within the journal author’s guidelines.

  1. Lines 15, 22, 27 and further in the text: Typhimurium is the serovar name, therefore it must be not italisized and must start with a capital letter (Salmonella Typhimurium).

Done.

  1. Line 59: I suggest adding subspecies here as follows: Salmonella enterica subsp. enterica serovar Typhimurium (S. Typhimurium)

Done.

  1. Lines 90, 94, and further in the text: lactobacilli should not be italisized

Done.

  1. Line 107: EIEC should not be italisized

Done.

  1. Line 116: Subscript as CO2

Done.

  1. Line 127: What is E.?

It was a typo, it has been removed

  1. Line 141: Add Roche for the LC Fast Start DNA Master SYBR Green kit

Done.

  1. Line 156: Add further details for ELISA: e.g. kit names etc.

Done.

  1. Line 190-191: Italisize Lactobacillus, L. brevis, L. rhamnosus

Done.

  1. Figures 1-4: Use better resolution for figures: text is not recognizable.

Done.

  1. Line 201: Italisize L. reuteri

Done.

  1. Figure 3: The right side of the figure doesn't fit.

The figures and their layout have been changed.

  1. Line 220: Italisize L.reuteri, L.rhamnosus and L. brevis

Done.

  1. Line 235: What is 540?

It’s a typo that was removed.

  1. Lines 323-324: Add Institutional Review Board Statement and Informed Consent Statement

The Institutional Review Board Statement and Informed Consent Statement are not required in our in vitro study.

Reviewer 3 Report

See the attached file for comments.

Author Response

Dear Reviewer, thank you for your careful revision of our manuscript. You can find our answers and corrections in attached file.

Round 2

Reviewer 1 Report

I appreciate the swift and accurate response to my inquiries. 

Author Response

Dear Reviewer,

thank you very much for appreciating our edits and our answers to your questions

We hope that the revised manuscript is now suitable for publication.

Reviewer 3 Report

The manuscript has been improved. Latin names should be revised, and lower-/uppercases checked.

As for the sentence in L329-330, if the authors intend that Lactobacilli are the genus with the highest relative or absolute abundance, it is not true, as it can be easily verified. If the meaning of that sentence is different, it should be rephrased.

Author Response

Dear Reviewer,

the recommended corrections to the text were all performed and hightlited.

We hope that the revised manuscript is now suitable for publication.
